# Effect of Temperature Cycling Pretreatment on the Thermal Stability of Sm_2_(Co, Fe, Zr, Cu)_17_ Magnets in the Mild Temperature Range

**DOI:** 10.3390/ma15248830

**Published:** 2022-12-10

**Authors:** Hulin Wu, Zhimei Long, Zhongsheng Li, Kaiqiang Song, Chaoqun Li, Dalong Cong, Bin Shao, Xiaowei Liu, Jianchun Sun, Yilong Ma

**Affiliations:** 1Southwest Institute of Technology and Engineering, Chongqing 400039, China; 2School of Metallurgy and Material Engineering, Chongqing University of Science and Technology, Chongqing 401331, China

**Keywords:** SmCo, thermal stability, magnet, temperature cycling

## Abstract

The irredeemable magnetic losses of Sm(Co, Fe, Zr, Cu)_7.8_ permanent magnets caused by oxidation are very important for their practical application. In this work, the simulated results with *R*^2^ ≥ 98% based on the data of the temperature cycling test and the long-term isothermal test for the original samples confirmed that the magnetic flux losses reached 9.38% after the 5000th cycle in range R.T.–300 °C, and 7.15% after oxidated at 180 °C for 10 years, respectively. Demagnetization curves showed that the low-temperature oxidation mainly led to the remanence attenuation, while the coercivity remained relatively stable. SEM observation and EDS analysis revealed that an oxide outer layer with a thickness of 1.96 μm was formed on the surface of the original sample at 180 °C for 180 days, in which there was no enrichment or precipitation of metal elements. However, once a Cu, O-rich outer layer with a thickness of 0.72 μm was grown by using a temperature cycling from −50–250 °C for three cycles, the attenuation of magnetic properties could be inhibited under the low-temperature oxidation. This work suggested that the magnetic attenuation of Sm_2_Co_17_-type permanent magnets in the low-temperature field could not be ignored, and provided a simple method to suppress this attenuation of magnetic properties below 300 °C.

## 1. Introduction

Heating a permanent magnet to higher temperatures causes not only irreversible losses of magnetization but also changes in the microstructure and the chemical composition which are not recoverable by magnetization [1,2,3,4]. The thermal stability of the magnetization of a permanent magnet at elevated temperatures is quite important in designing permanent magnet devices [5].

Rare-earth permanent magnets based on NdFeB or SmCo have relatively high energy densities, therefore, have applications in efficient motors and generators with high power-to-weight ratios [6,7]. NdDyFeB magnets with a Curie temperature ~320 °C can currently be operated at maximum temperatures of 150–180 °C [8,9]. Compared with NdFeB magnets, Sm_2_Co_17_-type magnets with a general formula of Sm(Co, Fe, Zr, Cu)_7.0–8.5_ exhibit more resistance to oxidation and corrosion, and are favored for the application of high temperatures up to 500 °C [10]. Therefore, Sm_2_Co_17_ magnets are the preferred choice for manufacturing permanent magnetic components exposed to harsh environments and higher temperatures.

During a long-term high-temperature application in an atmospheric environment, oxidation is the main factor causing the attenuation of magnetic properties [11,12,13,14], and the oxidation behavior is closely related to its microstructure [15,16]. Sm_2_Co_17_-type magnets exhibit a cellular structure with three phases: rhombohedral Fe-rich Sm_2_(Co, Fe)_17_ cellular phase (2:17R, R-3m) with Th_2_Zn_17_-type structure; hexagonal Cu-rich Sm(Co, Cu)_5_ cell boundary phase (1:5H, *P*_6_/mmm), and Zr-rich (Sm, Zr)Co_3_ lamellar phase [17]. The Zr-rich lamella phase with a thickness of 3 nm was reported to stabilize the cell structure and act as a diffusion path for Cu into the 1:5H cell boundary, leading to an increase in coercivity [18]. At temperatures > 500 °C in air, an external oxidation layer (EOS) composed of CuO_x_, Co_3_O_4_, and CoFe_2_O_4_ from the outer surface inwards was formed [13]. The wrinkling and buckling of the EOS caused heavier breakaway oxidation and a formation of a large volume fraction of α-Fe_2_O_3_ in the (Co_x_Fe_1−x_)Fe_2_O_4_ layer [14]. Cu outward diffusion through the inner (Co_x_Fe_1−x_)_3_O_4_ layer for the Cu_x_O growth was very slow, and the thickness almost remains unchanged [19]. Moreover, an internal oxidation zone (IZO) consisting of Sm oxides and CoFe matrixes was regarded as the primary cause of irredeemable magnetic loss [20], in which thickness increased with the oxidation time [21]. With the advancement of microstructure analysis technology, the understanding of the oxidation behavior of Sm_2_Co_17_ magnets at high temperatures is increasingly deepened, but the current means of preventing oxide formation at high temperatures are still mainly limited to the surface coating of magnets [22,23] or element substitution [24,25]. Although surface coatings have been shown to be very effective, this route may not be ideal when using these magnets on the rotor of an electrical machine, where they are often built into complex arrays with fine tolerances that require a final surface grind.

In the temperature range from room temperature to 180 °C, the remanence temperature coefficient of Sm_2_Co_17_ permanent magnets is below 0.05%·°C^−1^ in the temperature range from room temperature to 180 °C [26,27,28]. Zhang et al. reported that the temperature coefficients based on the losses of *B_r_* of Sm(Co, Fe, Zr, Cu)_7.8_ were 0.033%·°C^−1^ and 0.042%·°C^−1^ among the range of 20–100 °C and 100–200 °C, respectively [27]. However, for a long-term isothermal test, the losses of remanence (*B_r_*) and coercive force (*H_c__j_*) of Sm (Co, Fe, Zr, Cu)_6.93_ magnets after exposure of 200 h at room temperature in an atmospheric environment reached 7.0% and 13.9% [29]. Under the same exposure conditions, the losses of *B_r_* and *H_cj_* of Sm(Co, Fe, Zr, Cu)_7.60_ magnets were 15.7% and 9.3%, respectively [23]. Therefore, even in a low-temperature range below 300 °C, the magnetic temperature coefficients are not enough to evaluate the long-term magnetic attenuation performance of permanent magnets. We reported that the magnetic properties degradation and microstructure changes in Zn-coated NdFeB under a temperature cycling from room temperature to 180 °C, confirming that the oxidation behavior in the middle-low temperature range was quite different from that at high temperature; for example, the IZO layer was not observed [30]. As a result, there was only a loss of the remanence intensity, while the coercivity remained stable. As mentioned above, oxidation at high temperatures causes the wrinkling and buckling of EOS, so it was generally believed that EOS cannot prevent oxygen from entering the interior. Considering that the oxidation occurring at medium- and low-temperature ranges tend to be mild, the change in the EOS layer and its influence need to be further verified.

In this paper, a series of temperature cycling tests were carried out for the commercial Sm(Co, Fe, Zr, Cu)_7.8_ magnets with a limitation of the upper temperature ≤ 300 °C. Furthermore, a CuO_x_ outer layer on the surface of the Sm(Co, Fe, Zr, Cu)_7.8_ magnets was in-situ grown by a simple temperature cycling, and which effects magnetic properties attenuation and microstructure change were investigated during long-term isothermal tests.

## 2. Experiment and Characterization

### 2.1. Materials

Sintered bulk Sm(Co, Fe, Zr, Cu)_7.8_ magnets were obtained from Hubei Permanent Magnet Technology Co. Ltd. (Wuhan, China). The sample dimensions were 28 × 24 × 4.5 mm^3^, and the compositions of the sample are presented in Table 1. The residual magnetization strength (*B_r_*) of the original sample is 1.27 kGs, and the coercive force (*H_cj_*) of that is 20.39 kOe. The magnetic flux (*Φ*) is 20.38 mWb, and the surface magnetic field (*B*) is 191 mT.

### 2.2. High-Temperature Test

The High-temperature tests were carried out in a thermostatic furnace (electric drying box, 101, Zhongxing, Beijing, China). When the thermostatic furnace reached the set temperature, the sample was placed in the furnace hot zone at the required temperature (80, 120, 150, 180, 220, 250, 300, 350, 400, 450, 500, 550, 650, and 750 °C) ± 2 °C for 1 h. After insulation, the samples were cooled to room temperature in natural air. 

### 2.3. Temperature Cycling Test

For the temperature cycling test, high-temperature incubators (electric drying box, 101, Zhongxing) and low-temperature incubators (high and low-temperature laboratory, HL7005T, Yinhe, Chongqing, China) were used to reduce the influence of the heating and cooling times, respectively. All of the samples were first placed in the low-temperature incubator and kept at the set temperature for 1 h. Then, the samples were transferred to the high-temperature incubator within 1 min and also kept at the set temperature for 1 h. The entire process is defined as one cycle. The lower temperatures were set as −50 °C and room temperature (R.T.), while the upper limit temperatures were set as 180, 250, and 300 °C.

### 2.4. Long-Term Isothermal Test

In the long-term isothermal test, two kinds of samples were placed in a drying box (101, Zhongxing). The sample cycled from −50 °C to 250 °C for three cycles was defined as the pretreated sample, and the original sample was called an untreated sample. The temperature was set to 80 °C, 120 °C, and 180 °C, and the samples after 22, 45, 60, 90, 120, 150, and 180 days were taken for further characterization.

### 2.5. Characterization 

The microstructures of the Sm(Co, Fe, Zr, Cu)_7.8_ magnets were visualized by field emission scanning electron microscopy (FE-SEM, JEOL JSM-7800F, Tokyo, Japan). The elemental distribution maps and semi-quantitative chemical composition analyses were characterized by energy-dispersive X-ray spectrometry (EDS) associated with SEM using an X-MaxN 80 T detector (Oxford Instruments, Abingdon, UK). The magnetic flux (*Φ*) was measured at room temperature using a Maxwell meter (YC-820, Hunan, China) with Helmholtz coils (MM-150D, Hunan, China), and the Maxwell meter accuracy offset ± 10 μmT. The high and low-temperature laboratory (HL7005T, Yinhe, Chongqing, China) and the electric drying box (101, Beijing) were used to carry out the temperature cycling test and the long-term isothermal test. Demagnetization curves of the bulk samples and relevant magnetic parameters were tested at room temperature using a magnetograph (AMT-4, Shuangji, Sichuan, China), and the measurement error was less than ±2%. 

The average values calculated from three samples were used for each data point. Moreover, the losses of *Φ* (*Y*) were further defined as $Y=\frac{Y^{\prime }-Y_{0}}{Y_{0}}\;\times \;100\%$, where *Y′* and *Y*_0_ represent the *Φ* of the recycled and original samples, respectively.

## 3. Results and Discussion

### 3.1. Temperature Cycling Test

A high-temperature test was first performed to contrast with the results of the temperature cycling experiments. Figure 1 shows the values of *Φ* and its loss of the untreated Sm(Co, Fe, Zr, Cu)_7.8_ magnets from R.T. to 750 °C. As the temperature increases to 150, 180, and 300 °C, the *Φ* losses reach 2.88, 3.11, and 3.86% (Appendix A), respectively. The *Φ* losses remained stable in the temperature range from room temperature to 300 °C. When the exposure temperature exceeds 300 °C, the decay rate increases linearly with a large slope, indicating that the untreated Sm(Co, Fe, Zr, Cu)_7.8_ magnet has a relatively good thermal stability ≤ 300 °C.

The effect of the lower limit temperature on the *Φ* losses of the original untreated sample was first investigated by fixing the upper temperature at 180 °C during the temperature cycling. The differences in the *Φ* losses between −50–180 °C and R.T.–180 °C is only 0.46% after the 125th cycle (Appendix A), respectively. Therefore, the lower limit temperature was fixed at R.T. in the following experiments. Figure 2 shows the aging curves of the original Sm(Co, Fe, Zr, Cu)_7.8_ magnets cycled at different upper temperatures. All of the samples suffer the initial irreversible losses of 0.73, 0.95, and 1.24% after the 10th cycle, and then increase slowly to 1.67, 2.30, and 2.85% at 180, 250, and 300 °C after the 125th cycles (Appendix A), respectively. Even if the upper limit temperatures are lower than 300 °C, the original Sm(Co, Fe, Zr, Cu)_7.8_ magnets exhibit a gradual deterioration of *Φ* losses in the form of a power function as increasing the cycle number, which is different from the changing trend presented in Figure 1. Figure 3 presents the demagnetization curves and the magnetic properties of these untreated samples after the 125th cycle at different upper limit temperatures. At the upper limited temperatures of 180 °C and 300 °C, *B_r_* of oxidated samples are 1.02 T and 0.89 T with attention rates of 19.7% and 29.9%, and those *H_cj_* drop slightly to 20.03 kOe and 19.04 kOe with attention rates of 1.77% and 6.62%, respectively. The *B_r_* attenuation implies the damage of the orientation or integrity of the Sm_2_(Co, Fe)_17_ grains induced by oxidation [16]. Only a slight decrease in *H_cj_* indicates no significant change in the microstructure of the sample after oxidation. These results are similar to our previous study on the thermal ability of a Zn-coated NdFeB magnet [30].

The formula *Y = A ×* (*X − X_c_*)*^P^* is applied to fit the experimental data, where *Y* represents the losses, *X* is the cycle number, and *A*, *X_c_,* and *P* are the fitting parameters. The fitting results of these holding curves are listed in Table 2. The values of *P* keep in the range of 0.32–0.35, because the *P* is a magnetic viscosity coefficient which is mainly in connection with coercivity [31]. Increasing the upper limit temperature from 180 °C to 300 °C mainly leads to an increase in the *A* from 0.32 to 0.61. The values of *Φ* losses under the 1st, 3rd, 5th, and 5000th cycles are further simulated and shown in Table 3. The *Φ* losses from the low- to the high-temperature range are 0.47, 0.82, and 0.87% after the 3rd cycle, and are 0.56, 0.94, and 1.02% after the 5th cycle, indicating that the irreversible losses occur mainly in the first three cycles. After the 5000th cycle, the simulated values of *Φ* losses reach 6.44, 6.63, and 9.38%, respectively. Consistent with the results in Figure 1, the *Φ* losses are suddenly enhanced at 300 °C, but those at 180 °C and 250 °C are also too large enough to be neglected.

### 3.2. Long-Term Isothermal Test

The long-term isothermal test was used to simulate the continuous operation condition for the magnets. As mentioned above, a sample was pretreated by using temperature cycling from −50 °C to 250 °C for three cycles, compared with the untreated sample. The *Φ* losses of two types of samples after oxidation in air at 80, 120, and 180 °C for 180 days are shown in Figure 4. The *Φ* losses of the untreated sample under 180 days of oxidation are 2.72, 3.17, and 3.38%, and those of the pretreated sample are 2.43, 2.56, and 2.79% (Appendix A), respectively. The holding curves were also fitted by the formula of *Y = A ×* (*X − X_c_*)*^P^*, and the fitted parameters are given in Table 4. All of the curves exhibit good fitting results with *R*^2^ ≥ 0.98. As increasing the temperature from 80 °C to 180 °C, the *A* of untreated samples increases from 0.61 to 1.14, and the *P* decreases from 0.30 to 0.21. The simulated values of *Φ* losses under the 2000th, 3650th, and 6000th day were further calculated as shown in Table 5. After 3650 days (10 years) of oxidation at 80, 120, and 180 °C, the *Φ* losses of the untreated drop to 5.99, 6.52, and 7.15%, and those of the pretreated samples are 4.58, 5.08, and 5.39%, respectively. The inhibition effect of pretreatment on the attenuation of magnetic properties becomes more significant with the further extension of oxidation time.

Figure 5 shows the demagnetization curves and magnetic properties of the untreated original sample and pretreated sample before and after oxidization at 180 °C for 180 days. The *B_r_* of the untreated sample decreases from 1.27 kGs to 0.99 kGs, and that of the pretreated sample only decreases from 1.11 kGs to 1.05 kGs. The *H_cj_* of the untreated sample and the pretreated sample after oxidation decreased by 11.4% and 5.91%, respectively. Combined with the results shown in Table 4, this suggests that the *A* is more correlated with *B_r_*, and the *P* mainly depends on *H_cj_*. In addition, the changes in (*BH*)*_max_* and (e) *H_k_/H_cj_* after pretreatment are signified in Figure 5d,e. It notes that the (*BH*)*_max_* of drops by 49.0% and 9.7% after oxidated, compared with the (*BH*)*_max_* values of the fresh untreated sample and the fresh pretreated sample. The temperature cycling pretreatment indeed causes damage to the magnetic properties, but by which the magnetic properties of the sample declined are significantly inhibited under long-term oxidation.

Figure 6 shows the SEM images and the EDS results of the untreated samples before and after oxidization at 180 °C for 180 days. No obvious change between both SEM images was identified from the surface towards the matrix zone (Figure 6a,c). If the alloy sample consisted of a large number of metal oxides with worse conductivity, the SEM image should show a bright white area. EDS line profiles of the O element (Figure 6a,c) show that there is an outer oxide layer with a thickness of about 0.4 μm for the fresh untreated sample, and extend to 1.96 μm after 180 days. The oxide layer of the original sample should be formed during the storage step in an atmospheric environment. The EDS line spectra of Fe, Sm, Co, Cu, and Zr did not show a drastic concentration fluctuation. The EDS point analysis results (Figure 6b,d) confirm that the mole ratios of Sm, Co, Fe, Cu, and Zr are consistent with that of the original sample (Table 1), whether on the sample surface or inside the substrate. As a result, there is no EOS, IOZ, or transition oxidation zone in our case. It reported that at a higher holding temperature of 500 °C, the O element firstly diffused along the 1:5H boundary, resulting in further oxidation and decomposition of 1:3R, 1:5H, and 2:17R cells to form an IOZ containing soft magnetic compounds such as CoFe_2_O_4_, Fe_3_O_4_, and CoFe solid solution [17]. Due to this, the coercivity of the oxidized magnet exhibited a sharp attenuation [20,29]. Based on the above results, air exposure below 300 °C limits the oxidation of the magnetic phases to the outer layer.

For the pretreated samples, a bright gray outer layer with a thickness of 0.72 μm is formed after cycling from −50 °C to 250 °C for three cycles (Figure 7a), and which thickness extends to 1.28 μm after long-term oxidization (Figure 7c). EDS line profiles verify that different from the untreated samples, these bright gray layers of pretreated samples are rich in Cu and O (Figure 7a). Furthermore, the O contents in both Cu and O-rich outer layers formed before and after oxidation are only 4.40% and 4.87%, much lower than those of untreated samples (Figure 7b,d). This means that once a Cu, O-rich outer layer surface layer is formed, Cu atoms diffuse constantly from the inside to the surface and followed by oxidation, by which the oxidation of the magnetic main phase is weakened. Moreover, due to the relatively slow oxidation process at low temperatures, the CuO layer does not appear to crack and fracture. This is the reason why the magnetic performance degradation of the pretreated sample is suppressed. Similar to the results of the untreated sample, no metal oxides such as (CoFe)Fe_2_O_4_, Sm_2_O_3_, FeO_x_, or alloy compounds such as FeCo [14] are present in the pretreated sample, indicating that it is difficult for the decomposition of the main phase to occur below 300 °C.

## 4. Conclusions

The magnetic property attentions and microstructure changes in the original Sm(Co, Fe, Zr, Cu)_7.8_ magnet and that pretreated by using a temperature cycling from −50 °C to 250 °C for three cycles were extensively studied at a limited temperature below 300 °C. The conclusions were summarized as follows:(1)For temperature cycling in ranges of R.T.–180 °C and R.T.–300 °C, the *Φ* losses of the original samples after 125th cycles were 0.73% and 1.24%, respectively, corresponding to *B_r_* attenuation rates of 19.7% and 29.9%, and *H_cj_* attenuation rates of 1.77% and 6.62%.(2)The *Φ* losses of the original sample and the pretreated sample after oxidization at 180 °C reached 3.38% and 2.79% for 180 days, and further were simulated as 7.15% and 5.39% for 10 years. Meanwhile, *B_r_* and *H_cj_* at 180 °C for 180 days were 0.99 T and 18.07 kOe for the original sample and were 1.05 T and 18.37 kOe for the pretreated samples.(3)After the original sample oxidized at 180 °C for 180 days, there was only an oxide outer layer with a thickness of 1.96 μm without the enrichment or precipitation of metal elements. The Cu, O-rich outer layer with a thickness of 0.72 μm formed through the pretreatment, and further extended to 1.28 μm after oxidization for 180 days, by which point the attenuation of magnetic properties was effectively inhibited.

## Figures and Tables

**Figure 1 materials-15-08830-f001:**
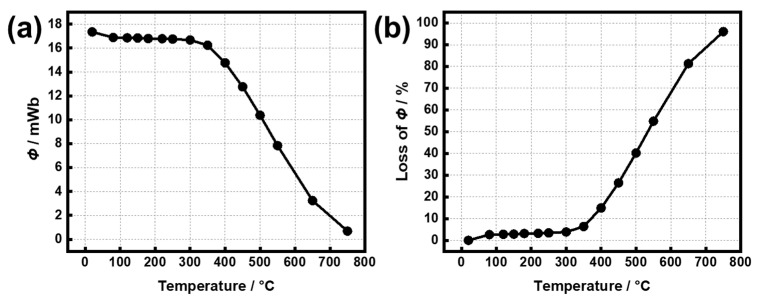
Temperature dependences of (**a**) *Φ* and (**b**) its losses of the untreated Sm(Co, Fe, Zr, Cu)_7.8_ samples from R.T. to 750 °C.

**Figure 2 materials-15-08830-f002:**
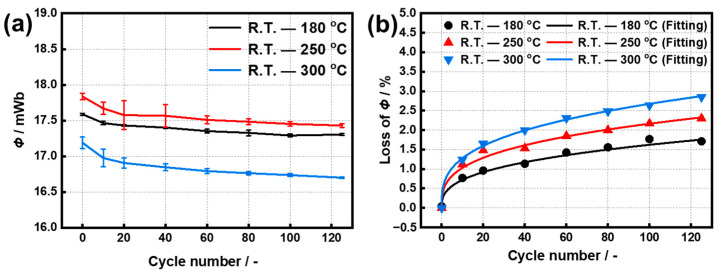
Variation trends of (**a**) *Φ*, and (**b**) the loss of *Φ* of the samples cycled in ranges of R.T. to 180, 250, and 300 °C, respectively. The line in (**b**) are the fitting curves with a formula *Y = A ×* (*X − X_c_*)*^P^*.

**Figure 3 materials-15-08830-f003:**
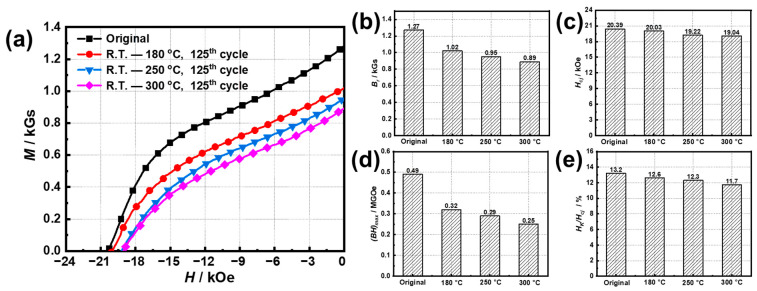
(**a**) Demagnetization curves, (**b**) *B_r_*, (**c**) *H_cj_*, (**d**) (*BH*)*_max_*, and (**e**) *H_k_/H_cj_* of the samples after the 125th cycle at different upper limit temperatures.

**Figure 4 materials-15-08830-f004:**
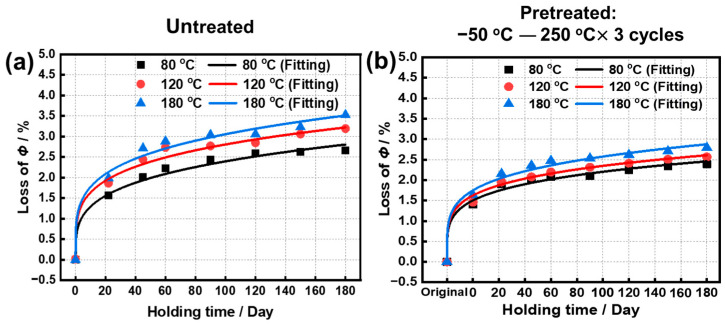
The *Φ* losses of (**a**) untreated and (**b**) pretreated samples after oxidation in air at 80, 120, and 180 °C for 180 days, respectively.

**Figure 5 materials-15-08830-f005:**
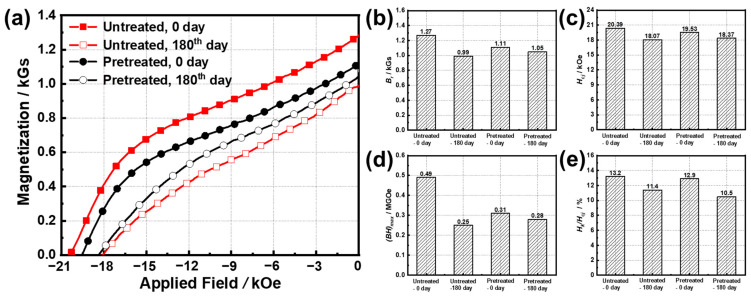
(**a**) Demagnetization curves, (**b**) *B_r_*, (**c**) *H_cj_*, (**d**) (*BH*)*_max_*, and (**e**) *H_k_/H_cj_* of the untreated and pretreated samples after oxidization in the air at 180 °C for 180 days.

**Figure 6 materials-15-08830-f006:**
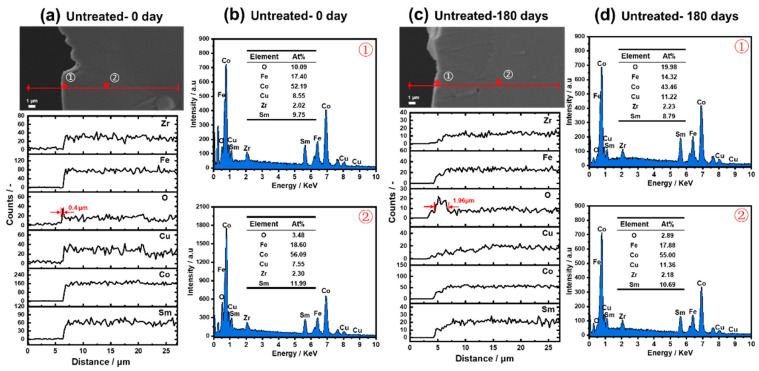
Cross-sectional backscattered SEM micrographs, EDS line profiles, and EDS point element analysis of different untreated samples. (**a**) SEM micrographs and EDS line profiles of the fresh untreated sample; (**b**) EDS point analysis on the outer layer and inside of the fresh untreated sample; (**c**) SEM micrographs and EDS line profiles of the untreated sample after oxidization at 180 °C for 180 days; (**d**) EDS point analysis on the outer layer and inside of the oxidized untreated sample.

**Figure 7 materials-15-08830-f007:**
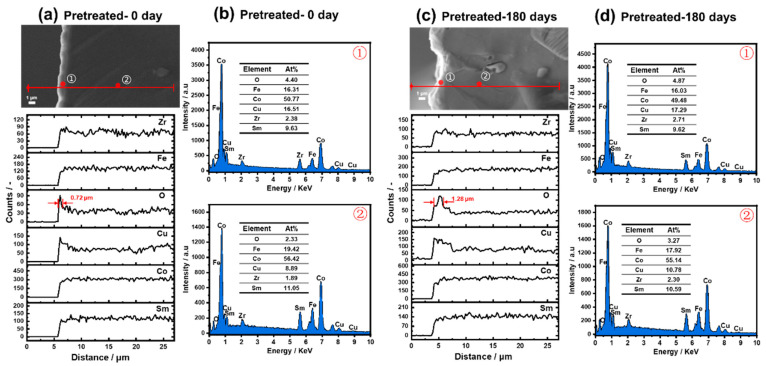
Cross-sectional backscattered SEM micrographs, EDS line profiles, and EDS point analysis of the pretreated samples before and after oxidization at 180 °C for 180 days. (**a**) SEM micrographs and EDS line profiles of the fresh pretreated sample, (**b**) EDS point analysis on the outer layer and inside of the fresh pretreated sample, (**c**) SEM micrographs and EDS line profiles of the pretreated sample after oxidization, and (**d**) EDS point element analysis on the outer layer and inside of the oxidized untreated sample.

**Table 1 materials-15-08830-t001:** Chemical compositions of the Sm(Co, Fe, Zr, Cu)_7.8_ sample used in this study (at.%).

Sample	Sm	Co	Fe	Cu	Zr
SmCo	11.43	57.37	19.03	9.94	2.23

**Table 2 materials-15-08830-t002:** Values of fitting parameters and their standard errors (*S.E.*) in the different temperature ranges. The fitting formulae are given by *Y = A ×* (*X − X_c_*)*^P^*.

*Y*	Temperature Range	Fitting Parameter						
		*X_c_*	*S.E.* _(*X*_ * _c_ * _)_	*A*	*S.E.* _(*A*)_	*P*	*S.E.* _(*P*)_	*R* ^2^
Loss of *Φ*	R.T.–180 °C	0.00	7.79	0.32	0.13	0.35	0.08	0.99
	R.T.–250 °C	0.00	8.38	0.49	0.20	0.32	0.09	0.98
	R.T.–300 °C	0.00	0.27	0.61	0.08	0.32	0.03	0.99

**Table 3 materials-15-08830-t003:** Simulation values of the losses of *Φ* in the different temperature range for the 1st, 3rd, 5th, and 5000th cycles, using the fitting parameters as shown in Table 2.

*Y*	Temperature Range	Simulated Value			
		1st Cycle	3rd Cycle	5th Cycle	5000th Cycle
Loss of *Φ*/%	R.T.–180 °C	0.32	0.47	0.56	6.44
	R.T.–250 °C	0.60	0.82	0.94	6.63
	R.T.–300 °C	0.61	0.87	1.02	9.38

**Table 4 materials-15-08830-t004:** Values of fitting parameters and their standard errors (*S.E.*) in the different temperature ranges. Fitting formulae are given by *Y = A ×* (*X − X_c_*)*^P^*.

Constant Temperature/°C	Pretreated	Fitting Parameter						
		*X_c_*	*S.E.* _(*Xc*)_	*A*	*S.E.* _(*A*)_	*P*	*S.E.* _(*P*)_	*R* ^2^
80	/	0.00	16.41	0.61	0.27	0.30	0.08	0.99
	−50–250 °C × 3 cycles	0.00	17.26	0.79	0.27	0.21	0.07	0.99
120	/	0.00	23.32	0.96	0.48	0.23	0.09	0.98
	−50–250 °C × 3 cycles	0.00	7.84	0.88	0.13	0.21	0.03	0.99
180	/	0.00	23.92	1.14	0.53	0.21	0.08	0.98
	−50–250 °C × 3 cycles	0.00	14.86	0.96	0.28	0.20	0.05	0.99

**Table 5 materials-15-08830-t005:** Simulation results of the losses of *Φ* in the different constant temperatures after the 2000th, 3650th, and 6000th days by using the fitting parameters as shown in Table 4.

Constant Temperature/°C	Pretreated	Loss of *Φ*/% (Simulated Value)		
		2000th Day	3650th Day	6000th Day
80	/	5.15	5.99	6.80
	−50–250 °C × 3 cycles	4.03	4.58	5.09
120	/	5.66	6.52	7.32
	−50–250 °C × 3 cycles	4.43	5.08	5.76
180	/	6.20	7.15	8.04
	−50–250 °C × 3 cycles	4.74	5.39	6.00

## Data Availability

The data used to support the findings of this study are available from the corresponding author upon request.

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
