# Peer review of "Effect of Temperature Cycling Pretreatment on the Thermal Stability of Sm2(Co, Fe, Zr, Cu)17 Magnets in the Mild Temperature Range"

_materials, 2022, doi:10.3390/ma15248830_

Round 1
Reviewer 1 Report
I reviewed the article Effect of temperature cycling pretreatment on the thermal stability of Sm2(Co, Fe, Zr, Cu)17 magnets in the mild temperature range. I have the following comments for the authors.
1. The Abstract is not acceptable, it is too short to communicate the article's overview to the readers. It should be thoroughly revised and enhanced giving a brief summary of materials-methods and results.
2. The literature review is not state-of-the-art. Out of about 26 references, I found no article from 2021, hardly 2 articles are from 2022. The literature review should be revised by including the most recent articles.
3. The Introduction is too short to identify the gaps in the literature and formulate the problem. It should be enhanced giving the recent developments in the compositions, and thermal performance of magnets in different application areas.
4. The Experimental methods (temperature cycling, high-temperature test, missing) need to introduce the test standards, the equipment, and their capacity, errors, along with the test environment and assumptions.
5. While the results are presented well, the discussion part is very short, there is no reference or comparison to the literature and applications. Particularly, figures 6 and 7 are the main outcomes of the article, hence relevant in-depth discussion is needed.
6. The SEM pictures are very small, they should be presented (or split) as separate pictures with large sizes and annotations.
7. Again conclusions should start from the summary of the research design and then present the main outcomes.
8. The temperature limitations (safe temperature) due to the thermal stability performance of the magnets and the potential impact of applications should be included in the conclusions.
Author Response
Thanks for the reminder, we've made changes to your comment, which will be uploaded via the attachment form. Thanks again for your suggestion.

Reviewer 2 Report
The manuscript "Effect of temperature cycling pretreatment on the thermal stability of Sm2(Co, Fe, Zr, Cu)17 magnets in the mild temperature range" is a very good manuscript, clear and exhaustive.
I have a minor comment: can the authors measure the hysteresis cycle in order to find the coercive field and residual magnetization..
Author Response

(The authors gave the same response as above.)
